# MULTIPOLAR: Multi-Source Policy Aggregation for Transfer Reinforcement Learning between Diverse Environmental Dynamics

## Abstract

Transfer reinforcement learning (RL) aims at improving learning efficiency of an agent by exploiting knowledge from other source agents trained on relevant tasks. However, it remains challenging to transfer knowledge between different environmental dynamics without having access to the source environments. In this work, we explore a new challenge in transfer RL, where only a set of source policies collected under unknown diverse dynamics is available for learning a target task efficiently. To address this problem, the proposed approach, MULTI-source POLicy AggRegation (MULTIPOLAR), comprises two key techniques. We learn to aggregate the actions provided by the source policies adaptively to maximize the target task performance. Meanwhile, we learn an auxiliary network that predicts residuals around the aggregated actions, which ensures the target policy's expressiveness even when some of the source policies perform poorly. We demonstrated the effectiveness of MULTIPOLAR through an extensive experimental evaluation across six simulated environments ranging from classic control problems to challenging robotics simulations, under both continuous and discrete action spaces.

## 1 Introduction

We envision a future scenario where a variety of robotic systems, which are each trained or manually engineered to solve a similar task, provide their policies for a new robot to learn a relevant task quickly. For example, imagine various pick-and-place robots working in factories all over the world. Depending on the manufacturer, these robots will differ in their kinematics (*e.g.*, link length, joint orientations) and dynamics (*e.g.*, link mass, joint damping, friction, inertia). They could provide their policies to a new robot (Devin et al., 2017), even though their dynamics factors, on which the policies are implicitly conditioned, are not typically available (Chen et al., 2018). Moreover, we cannot rely on a history of their individual experiences, as they may be unavailable due to a lack of communication between factories or prohibitively large dataset sizes. In such scenarios, we argue that a key technique to develop is the ability to transfer knowledge from a collection of robots to a new robot quickly *only by exploiting their policies while being agnostic to their different kinematics and dynamics*, rather than collecting a vast amount of samples to train the new robot from scratch.

The scenario illustrated above poses a new challenge in the transfer learning for reinforcement learning (RL) domains. Formally, consider multiple instances of a single environment that differ in their state transition dynamics, *e.g.*, independent ant robots with different leg designs in Figure 1, which reach different locations by executing the same walking actions. These source agents interacting with one of the environment instances provide their deterministic policy to a new target agent in another environment instance. Then, our problem is: *can we efficiently learn the policy of a target agent given only the collection of source policies?* Note that information about source environmental dynamics, such as the exact state transition distribu-

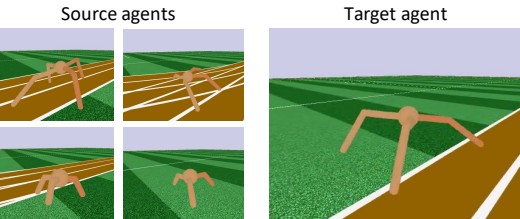

Figure 1: **Ant Example**. A policy of a target agent (right) is learned by utilizing the policies of other source agents with different leg designs (left).

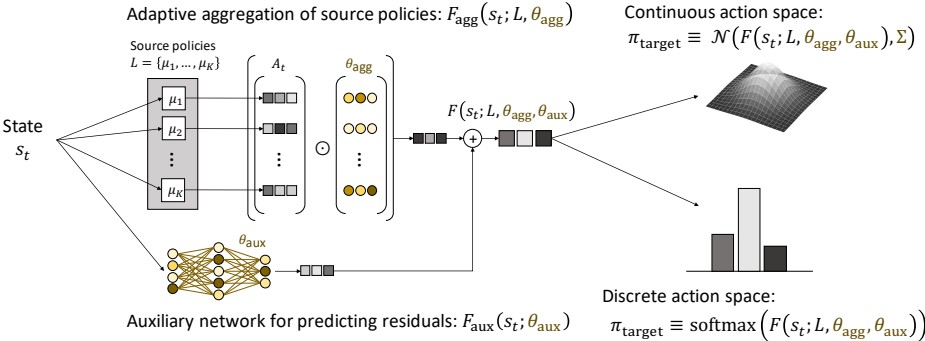

Figure 2: **Overview of MULTIPOLAR**. We formulate a target policy $\pi_{\text{target}}$ with the sum of 1) the adaptive aggregation $F_{\text{agg}}$ of deterministic actions from source policies $L$ and 2) the auxiliary network $F_{\text{aux}}$ for predicting residuals around $F_{\text{agg}}$.

tions and the history of environmental states, will not be visible to the target agent as mentioned above. Also, the source policies are neither trained nor hand-engineered for the target environment instance, and therefore not guaranteed to work optimally and may even fail (Chen et al., 2018). These conditions prevent us from adopting existing work on transfer RL between different environmental dynamics, as they require access to source environment instances or their dynamics for training a target policy (*e.g.*, Lazaric et al. (2008); Chen et al. (2018); Yu et al. (2019); Tirinzoni et al. (2018)). Similarly, meta-learning approaches (Vanschoren, 2018; Sæmundsson et al., 2018; Clavera et al., 2019) cannot be used here because they typically train an agent on a diverse set of tasks (*i.e.*, environment instances). Also, existing techniques that utilize a collection of source policies, *e.g.*, policy reuse frameworks (Fernández & Veloso, 2006; Rosman et al., 2016; Zheng et al., 2018) and option frameworks (Sutton et al., 1999; Bacon et al., 2017; Mankowitz et al., 2018), are not a promising solution because, to our knowledge, they assume source policies have the same environmental dynamics but have different goals.

As a solution to the problem, we propose a new transfer RL approach named **MULTI-source POLicy AggRegation (MULTIPOLAR)**. As shown in Figure 2, our key idea is twofold; 1) In a target policy, we adaptively aggregate the deterministic actions produced by a collection of source policies. By learning aggregation parameters to maximize the expected return at a target environment instance, we can better adapt the aggregated actions to unseen environmental dynamics of the target instance without knowing source environmental dynamics nor source policy performances. 2) We also train an auxiliary network that predicts a residual around the aggregated actions, which is crucial for ensuring the expressiveness of the target policy even when some source policies are not useful. As another notable advantage, the proposed MULTIPOLAR can be used for both continuous and discrete action spaces with few modifications while allowing a target policy to be trained in a principled fashion. Similar to Ammar et al. (2014); Song et al. (2016); Chen et al. (2018); Tirinzoni et al. (2018); Yu et al. (2019), our method assumes that the environment structure (state/action space) is identical between the source and target environments, while dynamics/kinematics parameters are different. This assumption holds in many real-world applications such as in sim-to-real tasks (Tan et al., 2018), industrial insertion tasks (Schoettler et al., 2019) (different dynamics comes from the differences in parts), and wearable robots (Zhang et al., 2017) (with users as dynamics).

We evaluate MULTIPOLAR in a variety of environments ranging from classic control problems to challenging robotics simulations. Our experimental results demonstrate the significant improvement of sample efficiency with the proposed approach, compared to baselines that trained a target policy from scratch or from a single source policy. We also conducted a detailed analysis of our approach and found it works well even when some of the source policies performed poorly in their original environment instance.

**Main contributions**: (1) a new transfer RL problem that leverages multiple source policies collected under diverse environmental dynamics to train a target policy in another dynamics, and (2) MULTIPOLAR, a simple yet principled and effective solution verified in our extensive experiments.

## 2    PRELIMINARIES

**Reinforcement Learning**    We formulate our problem under the standard RL framework (Sutton & Barto, 1998), where an agent interacts with its environment modeled by a Markov decision process (MDP). An MDP is represented by the tuple $\mathcal{M} = (\rho_0, \gamma, \mathcal{S}, \mathcal{A}, R, T)$ where $\rho_0$ is the initial state distribution and $\gamma$ is a discount factor. At each timestep $t$, given the current state $s_t \in \mathcal{S}$, the agent executes an action $a_t \in \mathcal{A}$ based on its policy $\pi(a_t \mid s_t; \theta)$ that is parameterized by $\theta$. The environment returns a reward $R(s_t, a_t) \in \mathbb{R}$ and transitions to the next state $s_{t+1}$ based on the state transition distribution $T(s_{t+1} \mid s_t, a_t)$. In this framework, RL aims to maximize the expected return with respect to the policy parameters $\theta$.

**Environment Instances**    In this work, we consider $K$ *instances of the same environment that differ only in their state transition dynamics.* We model each environment instance by an indexed MDP: $\mathcal{M}_i = (\rho_0, \gamma, \mathcal{S}, \mathcal{A}, R, T_i)$ where no two state transition distributions $T_i, T_j; i \neq j$ are identical. We also assume that each $T_i$ is unknown when training a target policy, *i.e.*, agents cannot access the exact form of $T_i$ nor a collection of states sampled from $T_i$.

**Source Policies**    For each of the $K$ environment instances, we are given a deterministic source policy $\mu_i : \mathcal{S} \to \mathcal{A}$ that only maps states to actions. Each source policy $\mu_i$ can be either parameterized (*e.g.*, learned from interacting with the environment modeled by $\mathcal{M}_i$) or non-parameterized (*e.g.*, heuristically designed by humans). Either way, we assume no prior knowledge about $\mu_i$ is available for a target agent, such as their representations or original performances, except that they were acquired in $\mathcal{M}_i$ with an unknown $T_i$.

**Problem Statement**    Given the set of source policies $L = \{\mu_1, \ldots, \mu_K\}$, our goal is to train a new target agent's policy $\pi_{\text{target}}(a_t \mid s_t; L, \theta)$ in a sample efficient fashion, where the target agent interacts with another environment instance $\mathcal{M}_{\text{target}} = (\rho_0, \mathcal{S}, \mathcal{A}, R, T_{\text{target}})$ and $T_{\text{target}}$ is not necessarily identical to $T_i$ ($i = 1 \ldots, K$).

## 3    MULTI-SOURCE POLICY AGGREGATION

As shown in Figure 2, with the Multi-Source Policy Aggregation (MULTIPOLAR), we formulate a target policy $\pi_{\text{target}}$ using a) the adaptive aggregation of deterministic actions from the set of source policies $L$, and b) the auxiliary network predicting residuals around the aggregated actions. We first present our method for the continuous action space, and then extend it to the discrete space.

**Adaptive Aggregation of Source Policies**    Let us denote by $a_t^{(i)} = \mu_i(s_t)$ the action predicted deterministically by source policy $\mu_i$ given the current state $s_t$. For the continuous action space, $a_t^{(i)} \in \mathbb{R}^D$ is a $D$-dimensional real-valued vector representing $D$ actions performed jointly in each timestep. For the collection of source policies $L$, we derive the matrix of their deterministic actions:

$$A_t = \left[ (a_t^{(1)})^\top, \ldots, (a_t^{(K)})^\top \right] \in \mathbb{R}^{K \times D}. \tag{1}$$

The key idea of this work is to aggregate $A_t$ adaptively in an RL loop, *i.e.*, to maximize the expected return. This adaptive aggregation gives us a "baseline" action that could introduce a strong inductive bias in the training of a target policy, without knowing source environmental dynamics $T_i$. More specifically, we define the adaptive aggregation function $F_{\text{agg}} : \mathcal{S} \to \mathcal{A}$ that produces the baseline action based on the current state $s_t$ as follows:

$$F_{\text{agg}}(s_t; L, \theta_{\text{agg}}) = \frac{1}{K} \mathbb{1}^K (\theta_{\text{agg}} \odot A_t), \tag{2}$$

where $\theta_{\text{agg}} \in \mathbb{R}^{K \times D}$ is a matrix of trainable parameters, $\odot$ is the element-wise multiplication, and $\mathbb{1}^K$ is the all-ones vector of length $K$. $\theta_{\text{agg}}$ is neither normalized nor regularized, and can scale each action of each policy independently. This means that we do not merely adaptively interpolate action spaces, but more flexibly emphasize informative source actions while suppressing irrelevant ones.

**Predicting Residuals around Aggregated Actions**    Moreover, we learn auxiliary network $F_{\text{aux}} : \mathcal{S} \to \mathcal{A}$ jointly with $F_{\text{agg}}$, to predict residuals around the aggregated actions. $F_{\text{aux}}$ is used to improve the target policy training in two ways. 1) If the aggregated actions from $F_{\text{agg}}$ are already useful in

the target environment instance, $F_{\mathrm{aux}}$ will correct them for a higher expected return. 2) Otherwise, $F_{\mathrm{aux}}$ learns the target task while leveraging $F_{\mathrm{agg}}$ as a prior to have a guided exploration process. Any network could be used for $F_{\mathrm{aux}}$ as long as it is parameterized and fully differentiable. Finally, the MULTIPOLAR function is formulated as:

$$F(s_t; L, \theta_{\mathrm{agg}}, \theta_{\mathrm{aux}}) = F_{\mathrm{agg}}(s_t; L, \theta_{\mathrm{agg}}) + F_{\mathrm{aux}}(s_t; \theta_{\mathrm{aux}}), \tag{3}$$

where $\theta_{\mathrm{aux}}$ denotes a set of trainable parameters for $F_{\mathrm{aux}}$. Note that the idea of predicting residuals for a source policy has also been presented by Silver et al. (2018); Johannink et al. (2019); Rana et al. (2019). The main difference here is that, while these works just *add* raw action outputs provided from a *single* hand-engineered source policy, we adaptively aggregate actions from multiple source policies in order to obtain a more flexible and canonical representation.

**Target Policy**   Target policy $\pi_{\mathrm{target}}$ can be modeled by reparameterizing the MULTIPOLAR function as a Gaussian distribution, *i.e.*, $\mathcal{N}(F(s_t; L, \theta_{\mathrm{agg}}, \theta_{\mathrm{aux}}), \Sigma)$, where $\Sigma$ is a covariance matrix estimated based on what the used RL algorithm requires. Since we regard $\mu_i \in L$ as fixed functions mapping states to actions, this Gaussian policy $\pi_{\mathrm{target}}$ is differentiable with respect to $\theta_{\mathrm{agg}}$ and $\theta_{\mathrm{aux}}$, and hence could be trained with any RL algorithm that explicitly updates policy parameters. Unlike Silver et al. (2018); Johannink et al. (2019); Rana et al. (2019), we can formulate the target policy in a principled fashion for actions in a discrete space. Specifically, instead of a $D$-dimensional real-valued vector, here we have a $D$-dimensional one-hot vector $a_t^{(i)} \in \{0, 1\}^D$, $\sum_j (a_t^{(i)})_j = 1$ as outputs of $\mu_i$, where $(a_t^{(i)})_j = 1$ indicates that the $j$-th action is to be executed. Following Eqs. (2) and (3), the output of $F(s_t; L, \theta_{\mathrm{agg}}, \theta_{\mathrm{aux}})$ can be viewed as $D$-dimensional un-normalized action scores, from which we can sample a discrete action after normalizing it by the softmax function.

## 4   EXPERIMENTAL EVALUATION

We aim to empirically demonstrate the sample efficiency of a target policy trained with MULTIPO-LAR (denoted by "MULTIPOLAR policy"). To complete the experiments in a reasonable amount of time, we set the number of source policies to be $K = 4$ unless mentioned otherwise. Moreover, we investigate the factors that affect the performance of MULTIPOLAR. To ensure fair comparisons and reproducibility of experiments, we followed the guidelines introduced by Henderson et al. (2018) and François-Lavet et al. (2018) for conducting and evaluating all of our experiments.

### 4.1   EXPERIMENTAL SETUP

**Baseline Methods**   To show the benefits of leveraging source policies, we compared our MULTI-POLAR policy to the standard multi-layer perceptron (MLP) trained from scratch, which is typically used in RL literature (Schulman et al., 2017; François-Lavet et al., 2018). As another baseline, we also used MULTIPOLAR with $K = 1$, which is an extension of residual policy learning (Silver et al., 2018; Johannink et al., 2019; Rana et al., 2019) (denoted by "RPL") with adaptive residuals as well as the ability to deal with both continuous and discrete action spaces. We stress here that the existing transfer RL or meta RL approaches that train a universal policy network agnostic to the environmental dynamics, such as Frans et al. (2018); Chen et al. (2018), cannot be used as a baseline since they require a policy to be trained on a distribution of environment instances, which is not possible in our problem setting. Also, other techniques using multiple source policies, such as policy reuse frameworks, are not applicable because their source policies should be collected under the target environmental dynamics.

**Environments**   To show the general effectiveness of the MULTIPOLAR policy, we conducted comparative evaluations of MULTIPOLAR on the following six OpenAI Gym environments: Ro-boschool Hopper, Roboschool Ant, Roboschool InvertedPendulumSwingUp, Acrobot, CartPole, and LunarLander. We chose these six environments because 1) the parameterization of their dynam-ics and kinematics is flexible enough, 2) they cover discrete action space (Acrobot and CartPole) as well as continuous action space, and 3) they are samples of three distinct categories of OpenAI Gym environments, namely Box2d, Classic Control, and Roboschool.

**Experimental Procedure**   For each of the six environments, we first created 100 environment instances by randomly sampling the dynamics and kinematics parameters from a specific range. For example, these parameters in the Hopper environment were link lengths, damping, friction,

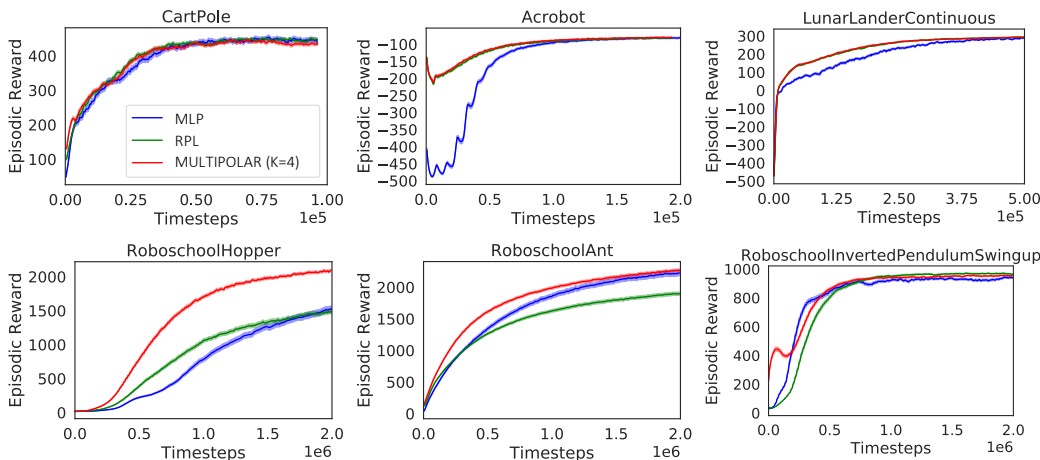

Figure 3: **Average Learning Curves** of MLP, RPL, and MULTIPOLAR ($K = 4$) over all the experiments for each environment. The shaded area represents 1 standard error.

armature, and link mass[1] Then, for each environment instance, we trained an MLP policy. The trained MLP policies were used in two ways: a) the baseline MLP policy for each environment instance, and b) a pool of 100 source policy candidates from which we sample $K$ of them to train MULTIPOLAR policies and one of them to train RPL policies[2]. Specifically, for each environment instance, we trained three MULTIPOLAR and three RPL policies with distinct sets of source policies selected randomly from the candidate pool. The learning procedure explained above was done three times with fixed different random seeds to reduce variance in results due to stochasticity. As a result, for each of the six environments, we had 100 environment instances $\times$ 3 random seeds $= 300$ experiments for MLP and 100 environment instances $\times$ 3 choices of source policies $\times$ 3 random seeds $= 900$ experiments for RPL and MULTIPOLAR. The aim of this large number of experiments is to obtain correct insights into the distribution of performances (Henderson et al., 2018). Due to the large number of experiments for all the environments, our detailed analysis and ablation study of MULTIPOLAR components were conducted with only Hopper, as its sophisticated second-order dynamics plays a crucial role in agent performance (Chen et al., 2018).

**Implementation Details**   All the experiments were done using the Stable Baselines (Hill et al., 2018) implementation of learning algorithms as well as its default hyperparameters and MLP network architecture for each environment (see Appendix A.1 for more details). Based on the performance of learning algorithms reported in the Hill et al. (2018), all the policies were trained with Soft Actor-Critic (Haarnoja et al., 2018) in the LunarLander environment and with Proximal Policy Optimization (Schulman et al., 2017) in the rest of the environments. For fair comparisons, in all experiments, auxiliary network $F_{\text{aux}}$ had an identical architecture to that of the MLP. Therefore, the only difference between MLP and MULTIPOLAR was the aggregation part $F_{\text{agg}}$, which made it possible to evaluate the contribution of transfer learning based on adaptive aggregation of source policies. Also, we avoided any random seed optimization since it has been shown to alter the policies' performance (Henderson et al., 2018).

**Evaluation Metric**   Following the guidelines of Henderson et al. (2018), to measure sampling efficiency of training policies, *i.e.*, how quick the training progresses, we used the average episodic reward over a various number of training samples. Also, to ensure that higher average episodic reward is representative of better performance and to estimate the variation of it, we used the sample bootstrap method (Efron & Tibshirani, 1993) to estimate statistically relevant 95% confidence bounds of the results of our experiments. Across all the experiments, we used 10K bootstrap iterations and the pivotal method. Further details on evaluation method can be found in Appendix A.3.

---

[1]Details of sampling ranges for dynamics and kinematics are provided in Appendix A.2.

[2]Although we used trained MLPs as source policies for reducing experiment times, any type of policies including hand-engineered ones could be used for MULTIPOLAR in principle.

Table 1: **MULTIPOLAR vs. Baselines**. Bootstrap mean and 95% confidence bounds of average episodic rewards over various training samples across six environments.

| Methods | CartPole | | | |
|---|---|---|---|---|
| | 25K | 50K | 75K | 100K |
| MLP | 171 (164,179) | 229 (220,237) | 266 (258,275) | 291 (282,300) |
| RPL | 185 (179,192) | 238 (231,245) | 269 (262,276) | 289 (282,296) |
| MULTIPOLAR (K=4) | **202 (195,209)** | **252 (245,260)** | **283 (276,290)** | **299 (292,306)** |
| | Acrobot | | | |
| | 50K | 100K | 150K | 200K |
| MLP | -305 (-317,-294) | -164 (-172,-156) | -127 (-133,-121) | -111 (-117,-106) |
| RPL | -154 (-159,-150) | -120 (-124,-116) | -105 (-109,-102) | -98 (-101,-95) |
| MULTIPOLAR (K=4) | **-151 (-155,-146)** | **-117 (-121,-113)** | **-103 (-106,-100)** | **-96 (-99,-93)** |
| | LunarLander | | | |
| | 125K | 250K | 375K | 500K |
| MLP | 10 (2,18) | 112 (104,121) | 178 (171,185) | 216 (210,221) |
| RPL | 92 (87,96) | 178 (174,182) | 223 (220,226) | 246 (243,248) |
| MULTIPOLAR (K=4) | **95 (90,99)** | **181 (177,185)** | **224 (221,228)** | **246 (244,249)** |
| | Roboschool Hopper | | | |
| | 0.5M | 1M | 1.5M | 2M |
| MLP | 26 (25,27) | 43 (42,45) | 67 (64,70) | 92 (88,96) |
| RPL | 37 (36,39) | 75 (70,79) | 114 (107,121) | 152 (142,160) |
| MULTIPOLAR (K=4) | **61 (59,64)** | **138 (132,143)** | **213 (206,221)** | **283 (273,292)** |
| | Roboschool Ant | | | |
| | 0.5M | 1M | 1.5M | 2M |
| MLP | 714 (674,756) | 1088 (1030,1146) | 1332 (1267,1399) | 1500 (1430,1572) |
| RPL | 807 (785,830) | 1120 (1088,1152) | 1307 (1269,1344) | 1432 (1391,1473) |
| MULTIPOLAR (K=4) | **1025 (995,1056)** | **1397 (1361,1432)** | **1606 (1568,1644)** | **1744 (1705,1783)** |
| | Roboschool InvertedPendulumSwingup | | | |
| | 0.5M | 1M | 1.5M | 2M |
| MLP | 159 (155,164) | 267 (260,273) | 347 (340,355) | 409 (401,417) |
| RPL | 111 (109,113) | 195 (192,198) | 265 (261,268) | 322 (317,326) |
| MULTIPOLAR (K=4) | **375 (355,395)** | **476 (456,495)** | **541 (522,559)** | **588 (571,605)** |

## 4.2 RESULTS

**Sample Efficiency of MULTIPOLAR** Figure 3 and Table 1 clearly show that on average, in all the environments, MULTIPOLAR outperformed baseline policies in terms of sample efficiency and sometimes the final episodic reward[3]. For example, in Hopper over 2M training samples, MULTI-POLAR with $K = 4$ achieved a mean of average episodic reward about three times higher than MLP (*i.e.*, training from scratch) and about twice higher than RPL (*i.e.*, using only a single source policy). It is also noteworthy that MULTIPOLAR with $K = 4$ had on par or better performance than RPL, which indicates the effectiveness of leveraging multiple source policies[4]. Figure 7 in Appendix, shows the individual average learning curve for each of the instances of Roboschool environments.

**Ablation Study** To demonstrate the importance of each component of MULTIPOLAR, we evaluated the following degraded versions: (1) $\theta_{\text{agg}}$ *fixed to 1*, which just averages the deterministic actions from the source policies without adaptive weights (similar to the residual policy learning

---

[3]Episodic rewards in Figure 3 are averaged over 3 random seeds and 3 random source policy sets on 100 environment instances. Table 1 reports the mean of this average over multiple numbers of training samples.

[4]Video replays of source policies, as well as MULTIPOLAR vs. baseline MLP in the Ant environment is available at: https://www.youtube.com/watch?v=3b0mGeT3sLo

Table 2: Results for MULTIPOLAR and its degraded versions in Hopper.

| MULTIPOLAR (K=4) | 0.5M | 1M | 1.5M | 2M |
|---|---|---|---|---|
| Full version | **61 (59,64)** | **138 (132,143)** | **213 (206,221)** | **283 (273,292)** |
| $\theta_{\mathrm{agg}}$ fixed to 1 | 56 (53,59) | 118 (111,126) | 180 (169,191) | 237 (222,250) |
| $F_{\mathrm{aux}}$ learned independent of $s_t$ | 53 (50,56) | 101 (95,108) | 146 (137,156) | 187 (175,200) |

Table 3: Results for MULTIPOLAR with different source policy sampling schemes in Hopper.

| MULTIPOLAR (K=4) | 0.5M | 1M | 1.5M | 2M |
|---|---|---|---|---|
| Random | 61 (59,64) | 138 (132,143) | 213 (206,221) | 283 (273,292) |
| 4 high performance | **98 (95,101)** | **214 (208,220)** | **323 (314,331)** | **420 (409,430)** |
| 2 high & 2 low performance | 45 (43,47) | 98 (94,102) | 154 (148,160) | 208 (200,215) |
| 4 low performance | 27 (26,27) | 45 (44,47) | 68 (66,71) | 92 (88,95) |

Table 4: Results for MULTIPOLAR with different number of source policies in Hopper.

| MULTIPOLAR | 0.5M | 1M | 1.5M | 2M |
|---|---|---|---|---|
| K=4 | 61 (59,64) | 138 (132,143) | 213 (206,221) | 283 (273,292) |
| K=8 | 71 (68,74) | 160 (154,167) | 246 (236,255) | 323 (312,335) |
| K=16 | **78 (75,80)** | **177 (172,182)** | **272 (264,279)** | **357 (348,367)** |

methods that used raw action outputs of a source policy), and (2) $F_{\mathrm{aux}}$ *learned independent of* $s_t$, which replaces the state-dependent MLP with an adaptive "placeholder" parameter vector making actions just a linear combination of source policy outputs. As shown in Table 2, the full version of MULTIPOLAR significantly outperformed both of the degraded versions, suggesting that the adaptive aggregation and predicting residuals are both critical.

**Effect of Source Policy Performances** Figure 4 illustrates an example of the histogram of final episodic reward (average rewards of the last 100 training episodes) for the source policy candidates obtained in the Hopper environment. As shown in the figure, the source policies were diverse in terms of the performance on their original environment instances[5]. In this setup, we investigate the effect of source policies performances on MULTIPOLAR sample efficiency.

We created two separate pools of source policies, where one contained only high-performing and the other only low-performing source policies[6]. Table 3 summarizes the results of sampling source policies from these pools (4 high, 2 high & 2 low, and 4 low performances) and compares them to the original MULTIPOLAR (shown as 'Random') also reported in Table 1. Not surprisingly, MULTIPOLAR performed the best when all the source policies were sampled from the high-performance pool. However, we emphasize that such high-quality policies are not always available in practice, due to the variability of how they are learned or hand-crafted under their own environment instance. Figure 6 in Appendix B.1 illustrates that MULTIPOLAR can successfully learn to suppress the useless low-performing source policies.

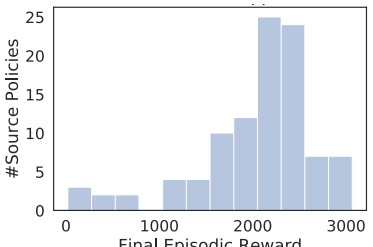

Figure 4: Histogram of source policy performances in Hopper.

**Effect of Number of Source Policies** Finally, we show how the number of source policies contributes to MULTIPOLAR's sample efficiency in Table 4. Specifically, we trained MULTIPOLAR policies up to $K = 16$ to study how the mean of average episodic rewards changes. The monotonic performance improvement over $K$ (for $K \leq 16$), is achieved at the cost of increased training and inference time. In practice, we suggest balancing this speed-performance trade-off by using as many source policies as possible before reaching the inference time limit required by the application.

---

[5]Histograms for the other environments can be found in Appendix A.4.

[6]Here, policies with final episodic reward over 2K are high-performing and below 1K are low-performing.

## 5    DISCUSSION AND RELATED WORK

Our work is broadly categorized as an instance of transfer RL (Taylor & Stone, 2009), in which a policy for a target task is trained using information collected from source tasks. In this section, we highlight how our work is different from the existing approaches and also discuss the current limitations as well as future directions.

**Transfer between Different Dynamics**    There has been very limited work on transferring knowledge between agents in different environmental dynamics. As introduced briefly in Section 1, some methods require training samples collected from source tasks. These sampled experiences are then used for measuring the similarity between environment instances (Lazaric et al., 2008; Ammar et al., 2014; Tirinzoni et al., 2018) or for conditioning a target policy to predict actions (Chen et al., 2018). Alternative means to quantify the similarity is to use a full specification of MDPs (Song et al., 2016; Wang et al., 2019) or environmental dynamics Yu et al. (2019). In contrast, the proposed MULTI-POLAR allows the knowledge transfer only through the policies acquired from source environment instances, which is beneficial when source and target environments are not always connected to exchange information about their environmental dynamics and training samples.

**Leveraging Multiple Policies**    The idea of utilizing multiple source policies can be found in the literature of policy reuse frameworks (Fernández & Veloso, 2006; Rosman et al., 2016; Li & Zhang, 2018; Zheng et al., 2018; Li et al., 2019). The basic motivation behind these works is to provide "nearly-optimal solutions" (Rosman et al., 2016) for short-duration tasks by reusing one of the source policies, where each source would perform well on environment instances with different rewards (*e.g.*, different goals in maze tasks). In our problem setting, where environmental dynamics behind each source policy are different, reusing a single policy without an adaptation is not the right approach, as described in (Chen et al., 2018) and also demonstrated in our experiment. Another relevant idea is hierarchical RL (Barto & Mahadevan, 2003; Kulkarni et al., 2016; Osa et al., 2019) that involves a hierarchy of policies (or action-value functions) to enable temporal abstraction. In particular, option frameworks (Sutton et al., 1999; Bacon et al., 2017; Mankowitz et al., 2018) make use of a collection of policies as a part of "options". However, they assumed all the policies in the hierarchy to be learned in a single environment instance. Another relevant work along this line of research is (Frans et al., 2018), which meta-learns a hierarchy of multiple sub-policies by training a master policy over the distribution of tasks. Nevertheless, hierarchical RL approaches are not useful for leveraging multiple source policies each acquired under diverse environmental dynamics.

**Learning Residuals in RL**    Finally, some recent works adopt residual learning to mitigate the limited performance of hand-engineered policies (Silver et al., 2018; Johannink et al., 2019; Rana et al., 2019). We are interested in a more extended scenario where various source policies with unknown performances are provided instead of a single sub-optimal policy. Also, these approaches focus only on RL problems for robotic tasks in the continuous action space, while our approach could work on both of continuous and discrete action spaces in a broad range of environments.

**Limitations and Future Directions**    Currently, our work has several limitations. First, MULTI-POLAR may not be scalable to a large number of source policies, as its training and testing times will increase almost linearly with the number of source policies. One possible solution for this issue would be pre-screening source policies before starting to train a target agent, for example, by testing each source on the target task and taking them into account in the training phase only when they are found useful. Moreover, our work assumes source and target environment instances to be different only in their state transition distribution. An interesting direction for future work is to involve other types of environmental differences, such as dissimilar rewards and state/action spaces.

## 6    CONCLUSION

We presented a new problem setting of transfer RL that aimed to train a policy efficiently using a collection of source policies acquired under diverse environmental dynamics. We demonstrated that the proposed MULTIPOLAR is, despite its simplicity, a principled approach with high training sample efficiency on a variety of environments. Our transfer RL approach is advantageous when one does not have access to a distribution of diverse environmental dynamics. Future work will seek to adapt our approach to more challenging domains such as a real-world robotics task.

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

## A  APPENDIX: FURTHER EXPERIMENTAL DETAILS

In this section, we present all the experimental details for the six environments we used. Note that we did not do any hyperparameter-tuning but followed the default parameters of Hill et al. (2018). We used the Roboschool implementation of Hopper, Ant, and InvertedPendulumSwingup since they are based on an open-source engine, which makes it possible for every researcher to reproduce our experiments. To run our experiments in parallel, we used GNU Parallel tool (Tange, 2018).

### A.1  HYPERPARAMETERS

Tables 5 and 6 summarize all the hyperparameters used for experiments on each environment. As done by Hill et al. (2018), to have a successful training, rewards and input observations are normalized using their running average and standard deviation for all the environments except CartPole and LunarLander. Also, in all of the experiments, $\theta_{\mathrm{agg}}$ is initialized to be the all-ones matrix.

Table 5: Hyperparameters for Acrobot, CartPole, Hopper, Ant and InvertedPendulumSwingup.

| PPO Parameters | Acrobot | CartPole | Hopper | Ant | InvertedPendulumSwingup |
|---|---|---|---|---|---|
| #Training samples | 200K | 100K | 2M | 2M | 2M |
| #Updates per rollout | 4 | 20 | 10 | 10 | 10 |
| Learning rate | 2.5e-4 | 1e-3 | 2.5e-4 | 2.5e-4 | 2.5e-4 |
| Mini batch size | 8 | 1 | 128 | 32 | 32 |
| Discount factor | 0.99 | 0.98 | 0.99 | 0.99 | 0.99 |
| GAE $\lambda$ | 0.94 | 0.8 | 0.95 | 0.95 | 0.95 |
| Clip ratio | 0.2 | 0.2 | 0.2 | 0.2 | 0.2 |
| Value function coefficient | 0.5 | 0.5 | 0.5 | 0.5 | 0.5 |
| Entropy coefficient | 0 | 0 | 0 | 0 | 0 |
| Gradient clipping value | 0.5 | 0.5 | 0.5 | 0.5 | 0.5 |
| Optimizer | Adam | Adam | Adam | Adam | Adam |
| MLP & $F_{\mathrm{aux}}$ Parameters | | | | | |
| Hidden layers | 64-64 | 64-64 | 64-64 | 16 | 64-64 |
| Activation functions | tanh | tanh | tanh | tanh | tanh |

Table 6: Hyperparameters for LunarLander.

| SAC Parameters | LunarLander |
|---|---|
| #Training samples | 500K |
| #Steps before learning starts | 1K |
| Buffer size | 50K |
| Learning rate | 3e-4 |
| Mini batch size | 256 |
| Discount factor | 0.99 |
| Soft update coefficient $\tau$ | 5e-3 |
| Entropy coefficient | learned automatically |
| Model training frequency | 1 |
| Target network training frequency | 1 |
| #Gradient updates after each step | 1 |
| Probability of taking a random action | 0 |
| Action noise | none |
| Optimizer | Adam |
| MLP & $F_{\mathrm{aux}}$ Parameters | |
| Hidden layers | 64-64 |
| Activation functions | relu |

## A.2    Sampling Range of the Environmental Parameters

Sampling ranges for dynamics and kinematics of each environment are provided in Tables 7, 8, 9, 10, 11 and 12. We defined these sampling ranges such that the resulting environments are stable enough for successfully training an MLP policy. To do so, we trained MLP policies across wide ranges of environmental parameters and chose the ranges in which the policy converged.

Table 7: Sampling range for Ant kinematic and dynamic parameters.

| Kinematics | |
|---|---|
| Links | Length Range |
| Legs | [0.4, 1.4] × default length |
| Dynamics | |
| Damping | [0.1, 5] |
| Friction | [0.4, 2.5] |
| Armature | [0.25, 3] |
| Links mass | [0.7, 1.1] × default mass |

Table 8: Sampling range for CartPole kinematic and dynamic parameters.

| Kinematics | |
|---|---|
| Links | Length Range (m) |
| Pole | [0.1, 3] |
| Dynamics | |
| Force | [6, 13] |
| Gravity | [-14, -6] |
| Poll mass | [0.1, 3] |
| Cart mass | [0.3, 4] |

Table 9: Sampling range for Hopper kinematic and dynamic parameters.

| Kinematics | |
|---|---|
| Links | Length Range (m) |
| Leg | [0.35, 0.65] |
| Foot | [0.29, 0.49] |
| Thigh | [0.35,0.55] |
| Torso | [0.3,0.5] |
| Dynamics | |
| Damping | [0.5, 4] |
| Friction | [0.5, 2] |
| Armature | [0.5, 2] |
| Links mass | [0.7, 1.1] × default mass |

Table 10: Sampling range for InvertedPendulumSwingup kinematic and dynamic parameters.

| Kinematics | |
|---|---|
| Links | Length Range (m) |
| Pole | [0.2, 2] |
| Dynamics | |
| Damping | [0.1, 5] |
| Friction | [0.5, 2] |
| Armature | [0.5, 3] |
| Gravity | [-11, -7] |
| Links mass | [0.4, 3] × default mass |

Table 11: Sampling range for Acrobot kinematic and dynamic parameters.

| Kinematics | |
|---|---|
| Links | Length Range (m) |
| Link 1&2 | [0.3, 1.3] |
| Dynamics | |
| Links mass | [0.5, 1.5] |
| Links center mass | [0.05, 0.95] × link length |
| Links inertia moments | [0.25, 1.5] |

Table 12: Sampling range for LunarLander kinematic and dynamic parameters.

| Kinematics | |
|---|---|
| Side engine height | [10, 20] |
| Dynamics | |
| Scale | [25, 50] |
| Initial Random | [500, 1500] |
| Main engine power | [10, 40] |
| Side engine power | [0.5, 2] |
| Side engine away | [8, 18] |

## A.3 EVALUATION METHOD

In this section, we explain how we calculated the mean of average episodic rewards in Tables 1, 2, 3, and 4, over a specific number of training samples (the numbers at the header of the tables *e.g.*, 25K, 50K, 75K, and 100K for the CartPole) which we denote by $T$ in what follows. For each experiment in an environment instance, we computed the average episodic reward by taking the average of the rewards over all the episodes the agent played from the beginning of the training until collecting $T$ number of training samples. Then we collected the computed average episodic rewards of all the experiments, *i.e.*, all the combinations of three random seeds, three random sets of source policies (for RPL and MULTIPOLAR), and 100 target environment instances. Finally, we used the sample bootstrap method (Efron & Tibshirani, 1993) to estimate the mean and the 95% confidence bounds of the collected average episodic rewards. We used the Facebook Boostrapped implementation: https://github.com/facebookincubator/bootstrapped.

## A.4 SOURCE POLICIES HISTOGRAMS

To generate environment instances, we uniformly sampled the dynamics and kinematics parameters from the ranges defined in Section A.2. Figure 5 illustrates the histograms of the final episodic rewards of source policies on the original environment instances in which they were acquired.

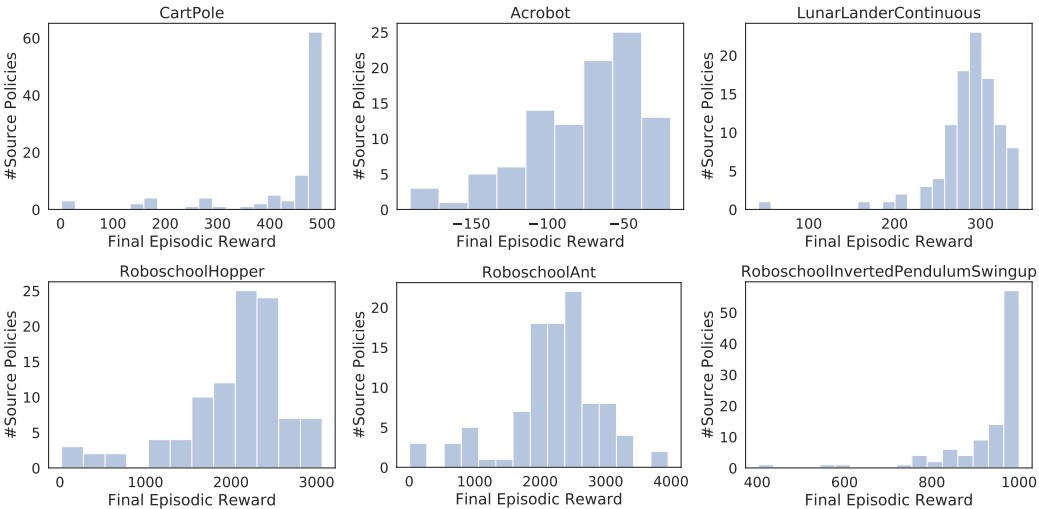

Figure 5: Histogram of final episodic rewards obtained by source policies per environment.

# B    APPENDIX: ADDITIONAL RESULTS

## B.1    LEARNED AGGREGATION PARAMETERS VISUALIZATION

Figure 6 visualizes an example of how the aggregation parameters $\theta_{\mathrm{agg}}$ for the four policies and their three actions were learned during the 2M timestep training of MULTIPOLAR ($K = 4$) policy in the Hopper environment. In this example, the source policies in the first and second rows were sampled from low-performance pools whereas those in the third and fourth rows were sampled from high-performance pools (see Section 4.2 for more details). It illustrates that MULTIPOLAR can successfully suppress the two useless low-performing policies as the training progresses.

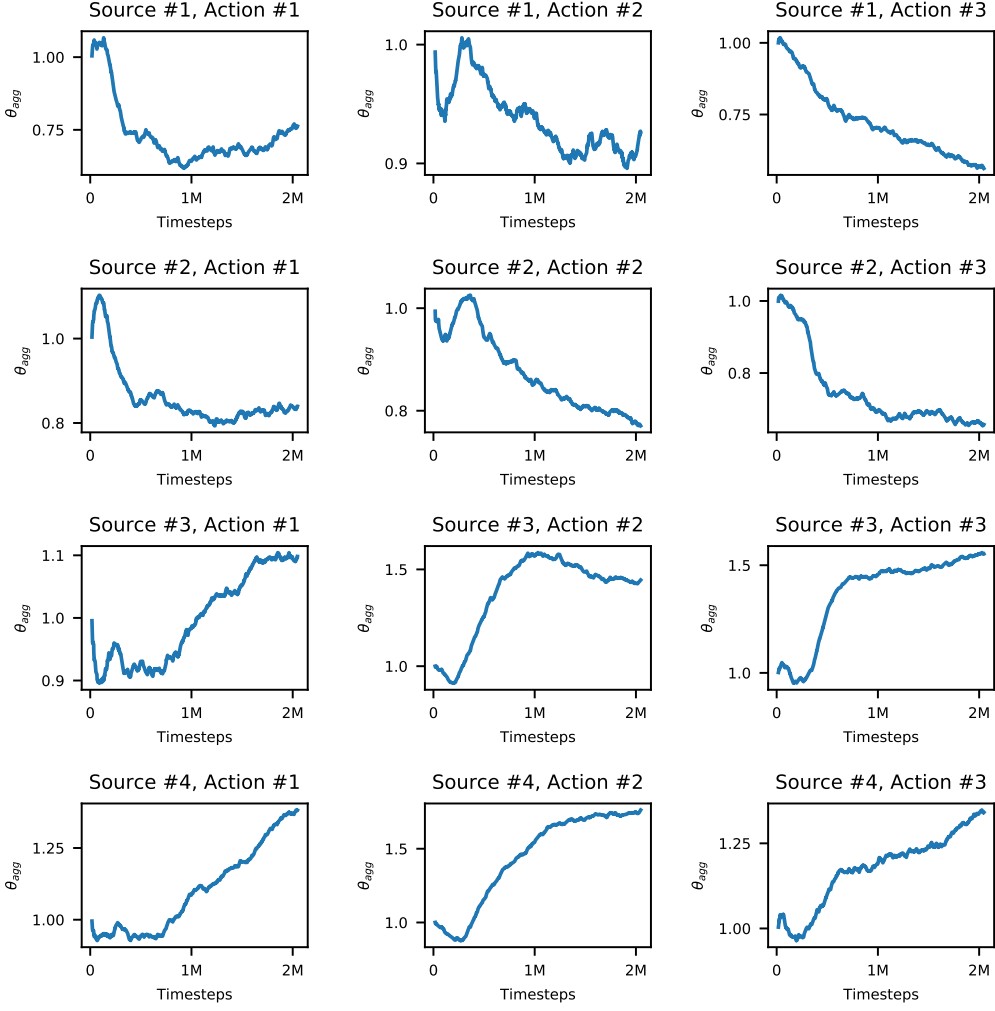

Figure 6: Aggregation parameters $\theta_{\mathrm{agg}}$ during the training of MULTIPOLAR ($K = 4$) in the Hopper that has 3-dimensional actions. Here, the first two source policies are low-performing and the last two are high-performing in their original environment instance.

## B.2    MULTIPOLAR WITH RANDOMLY INITIALIZED POLICIES

To further study how having low-performing source policies affects MULTIPOLAR sample efficiency, we evaluated MULTIPOLAR (K=4) in the Hopper environment, where the sources are randomly initialized policies, *i.e.*, policies that predict actions randomly. Following our experimental procedure explained in Section 4.1, Table 13 reports the bootstrap mean and 95% confidence bounds

of average episodic rewards over various training samples for this experiment and compares it with MULTIPOLAR with four low-performing sources. This result suggests that the sample efficiency of MULTIPOLAR ($K = 4$) with low-performing source policies (*i.e.*, source policies which had low final episodic rewards in their own environments) is almost the same as with randomly initialized source policies.

Table 13: Results for MULTIPOLAR with low-performing source policies vs. with randomly initialized source policies in Hopper.

| MULTIPOLAR (K=4) | 0.5M | 1M | 1.5M | 2M |
|---|---|---|---|---|
| 4 randomly initialized | 27 (26,28) | **47 (45,49)** | **73 (70,76)** | **101 (96,106)** |
| 4 low performance [Table 3] | 27 (26,27) | 45 (44,47) | 68 (66,71) | 92 (88,95) |

### B.3   INDIVIDUAL AVERAGE LEARNING CURVES

As an example, we visualized the individual average learning curve of policies for each of the environment instances of the Roboschool environments. Figures 7, 8 and 9 compare the individual average learning curves of MULTIPOLAR to the baseline policies in the 100 target environment instances.

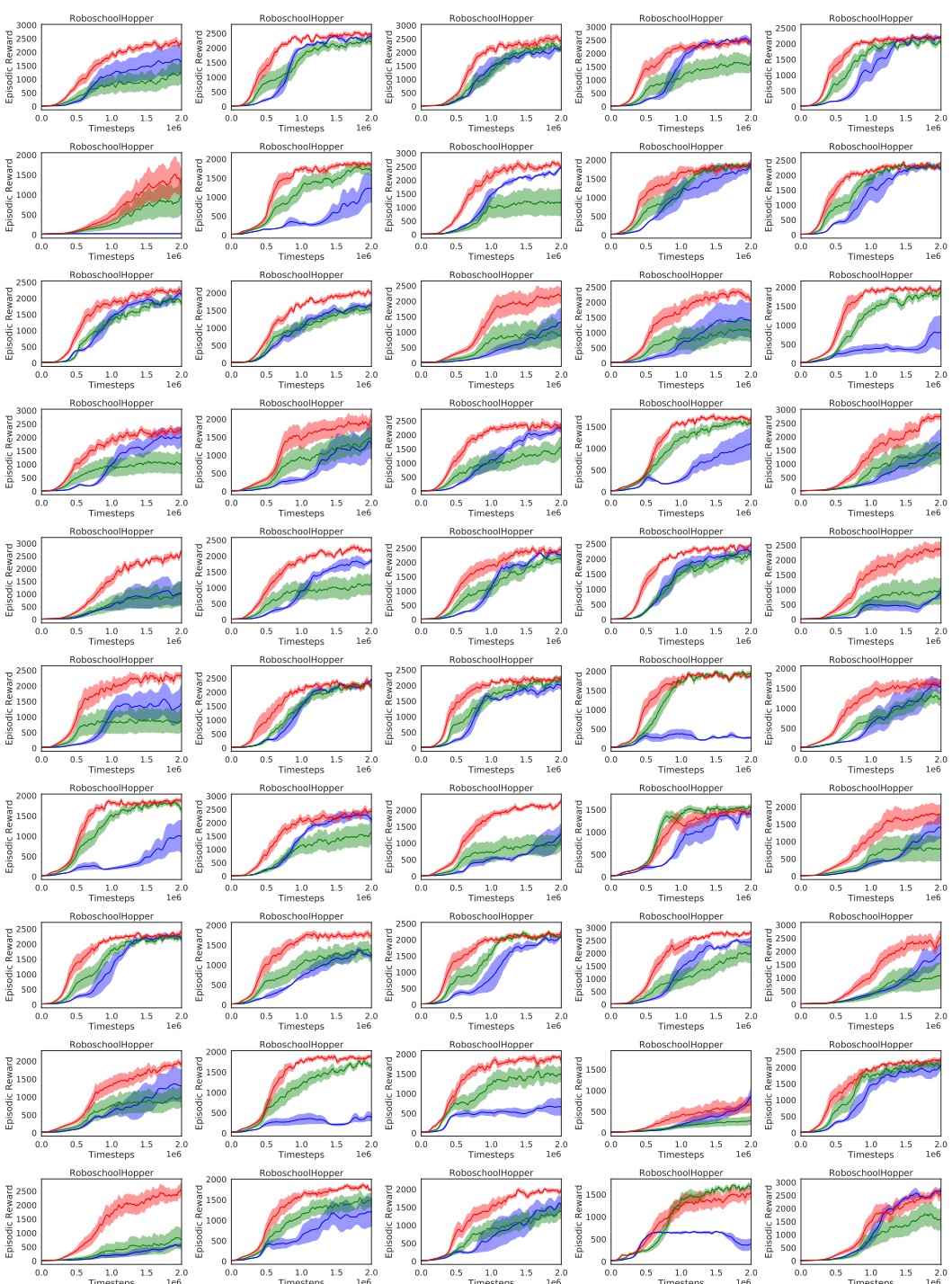

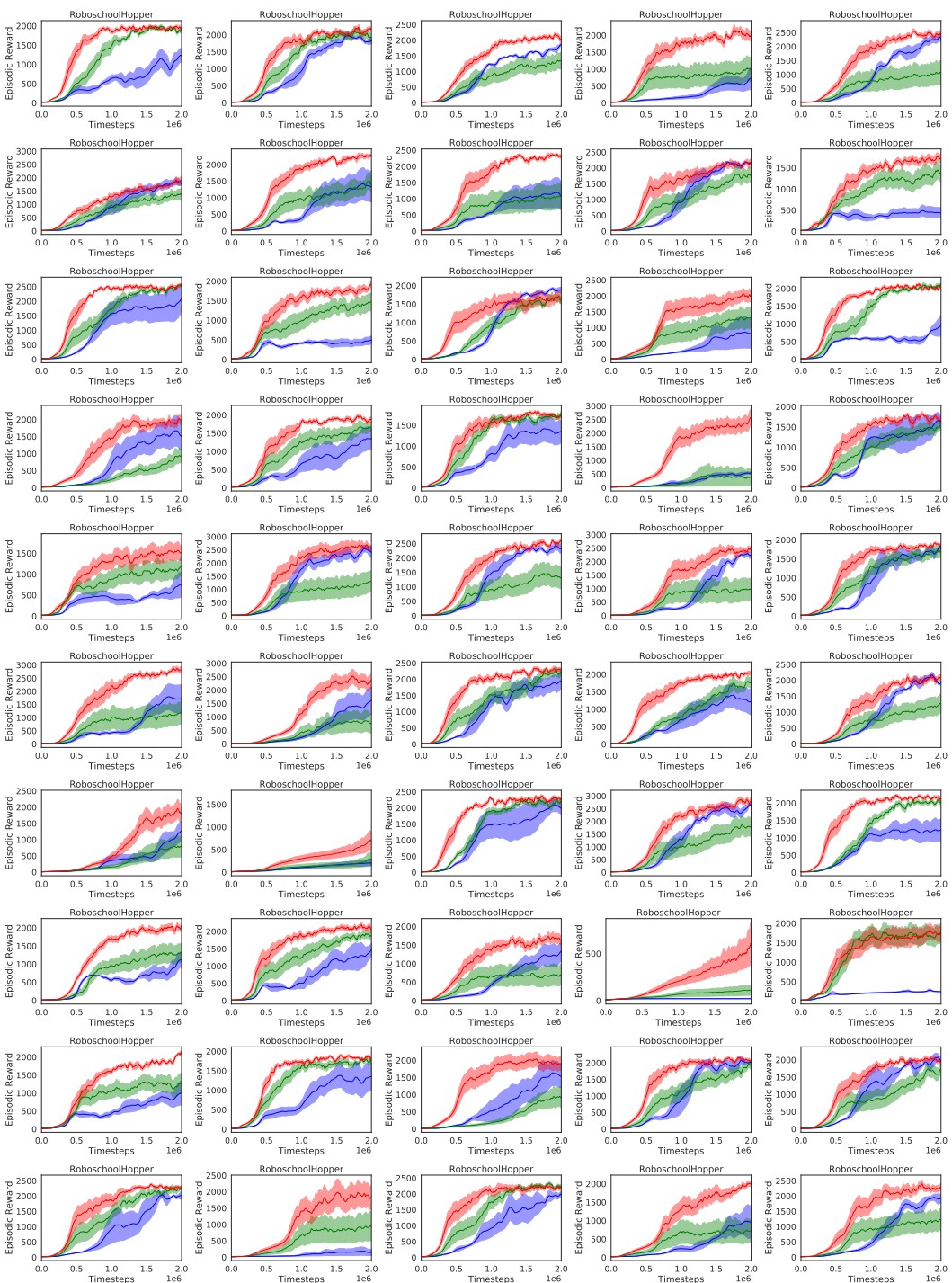

Figure 7: Average learning curves of MULTIPOLAR with $K = 4$ in red, RPL in green and MLP in blue over 3 random seeds and 3 random source policy sets for all the 100 target environment instances of Hopper.

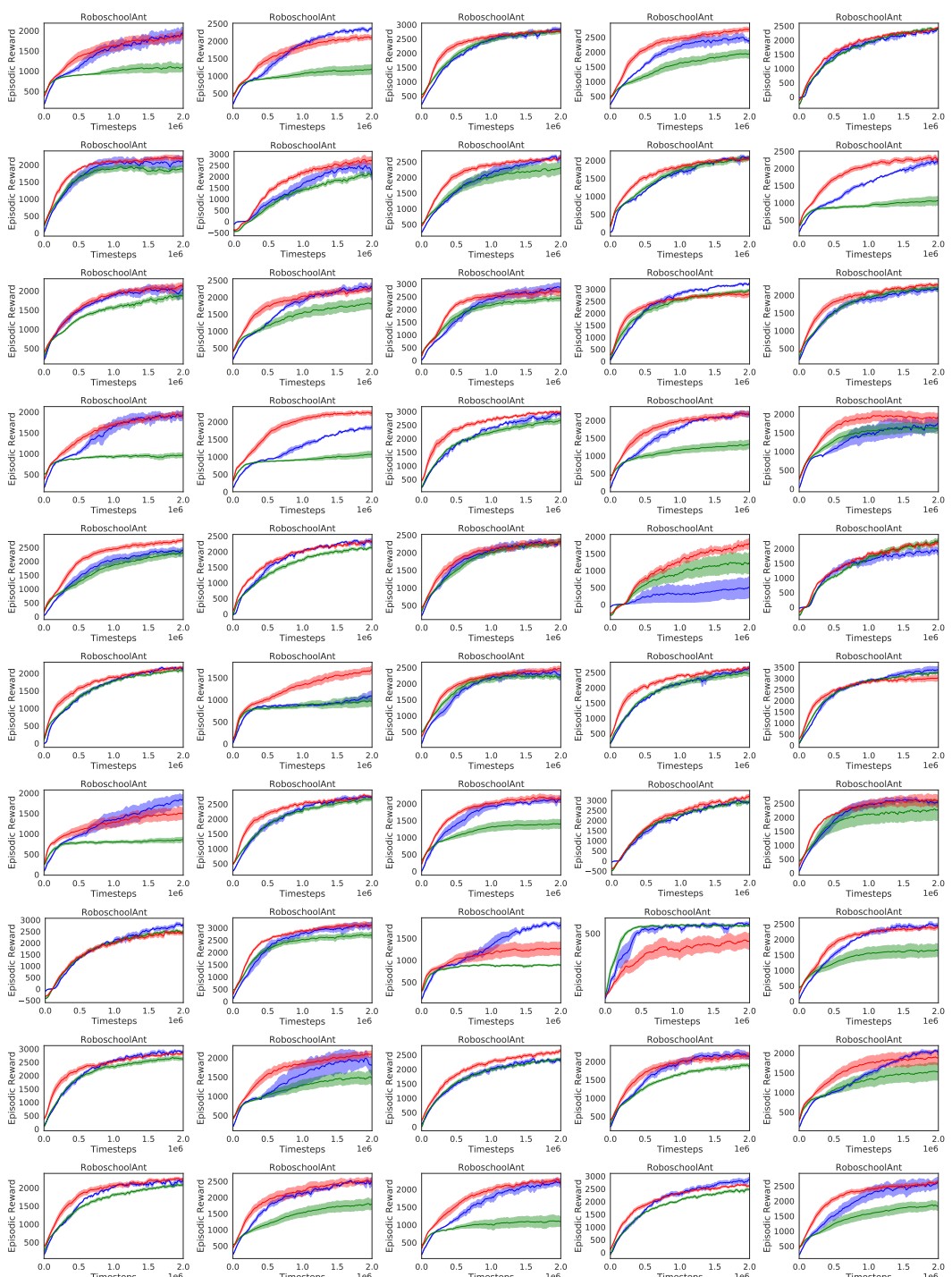

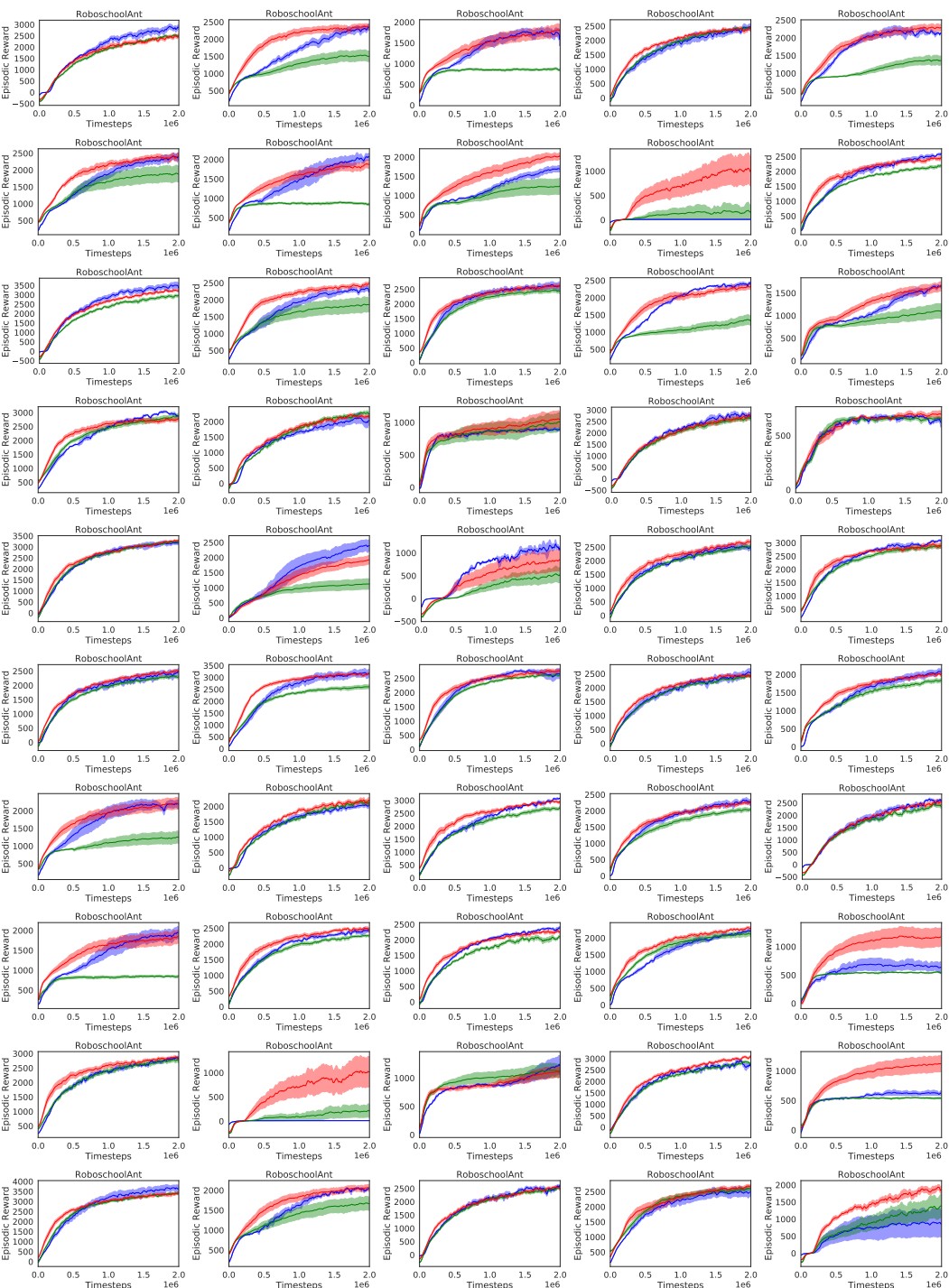

Figure 8: Average learning curves of MULTIPOLAR with $K = 4$ in red, RPL in green and MLP in blue over 3 random seeds and 3 random source policy sets for all the 100 target environment instances of Ant.

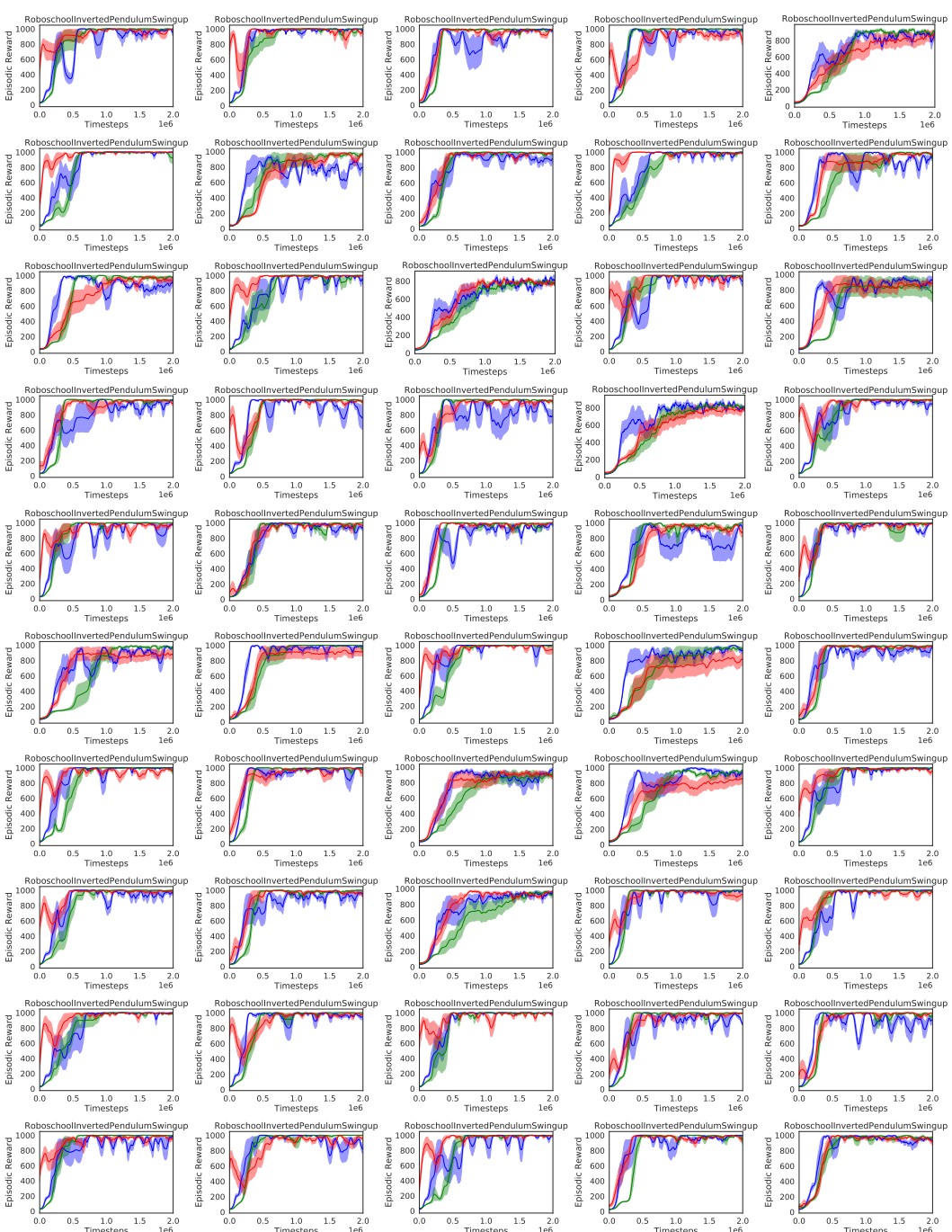

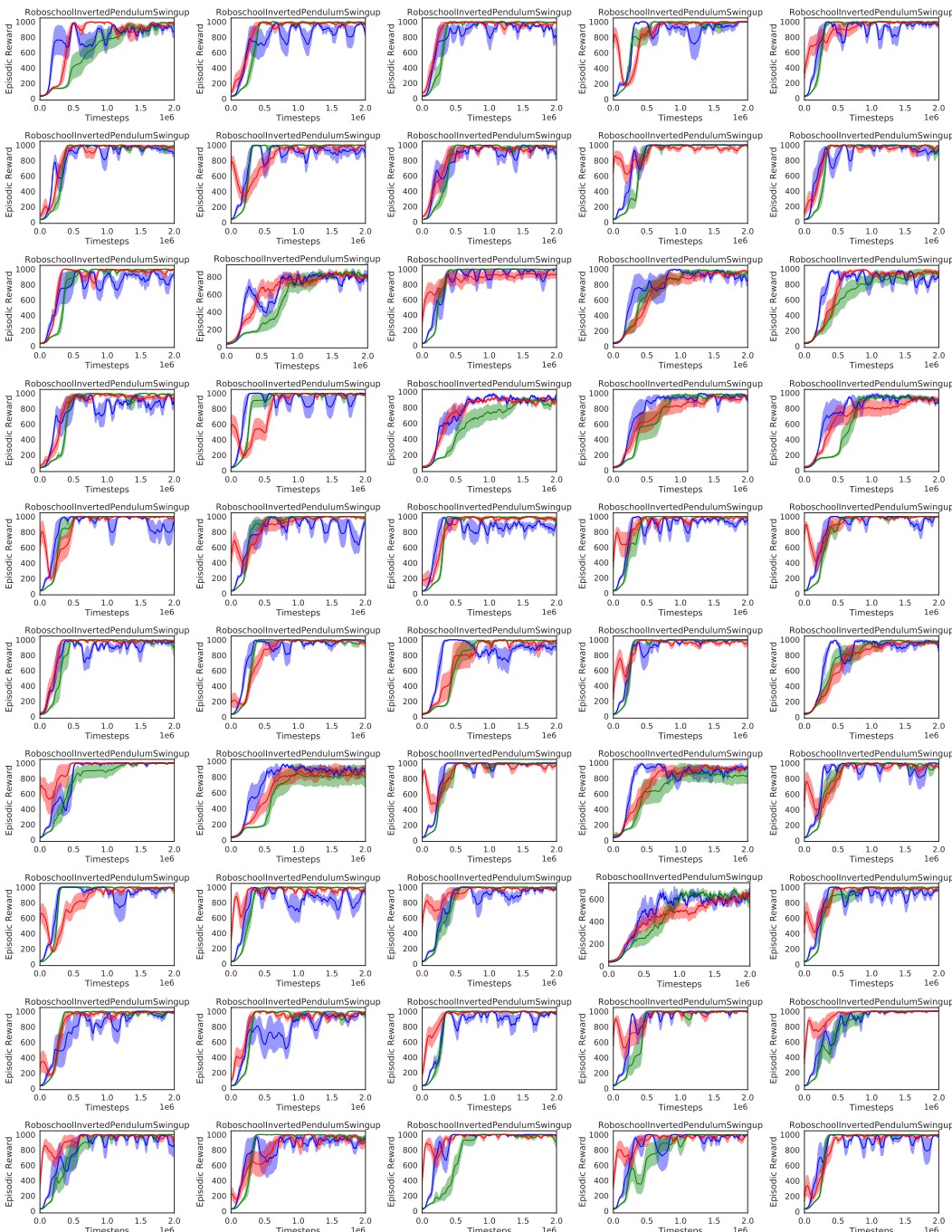

Figure 9: Average learning curves of MULTIPOLAR with $K = 4$ in red, RPL in green and MLP in blue over 3 random seeds and 3 random source policy sets for all the 100 target environment instances of InvertedPendulumSwingup.

