# OpenReview forum: "MULTIPOLAR: Multi-Source Policy Aggregation for Transfer Reinforcement Learning between Diverse Environmental Dynamics"
_ICLR.cc/2020/Conference — Reject_

### Official Review · AnonReviewer3 · 2019-10-09
**Official Blind Review #3**

**Rating:** 1

**Review:**

The paper introduces a novel multi-source policy transfer problem, where we want to utilize policies from multiple source domains with different dynamics to improve the performance of the policy on our target dynamics.

The paper addresses the problem by adaptively aggregating the deterministic actions produced by source policies to maximize the expected return in the target environment. The method further trains an auxiliary network to predict a residual to revise the predicted action when some source policies are not useful or even adversarial.

 In my understanding, the paper assumes that the source and target domain shared the same task (reward structure) but only differs in dynamics. Also, the two domains share similar state and action spaces since the policy accepts the target states and predict actions in the target environment. This may limit the usage of the method.

The paper proposes to use residual learning as an auxiliary to compensate for the sub-optimal expressiveness of the source policies, which is novel and interesting.

The paper performs experiments on multiple environments. But the source and target domains only vary in some parameters of the agents. The domain gap seems small for these experiments. The paper needs a metric to measure the domain gap between source and target dynamics and report how the domain gap influences the proposed method and the baselines according to the metric.

One of my major concerns is that the limitation of the method with the same state and action space of the source and target domains. Also, there is no theoretical or intuitive analysis of how large the domain gap can be. This problem can be impractical for real-world applications with restrict limitations.

Post-rebuttal:

My major concern is that the method is a naive combination of previous works and the paper is more like an engineering work. The method is also a weighted sum of source policies. There is no insight why the combination can work.

No assumption on source policies is given. That means I can get any random policy to learn a combination. This is like learning a policy from scratch by reinforcement learning. With better source policies, we can achieve better initialization for RL.

The paper is also related to hierarchical reinforcement learning, where the target reinforcement learning step is like building a high-level policy.

The work still requires lots of steps to train in the target domain, which does not fit to the real application of transfer RL. We hope transfer learning can adapt to the target environment fast.

**Experience Assessment:**

I have read many papers in this area.

**Review Assessment: Checking Correctness Of Derivations And Theory:**

I carefully checked the derivations and theory.

**Review Assessment: Checking Correctness Of Experiments:**

I carefully checked the experiments.

**Review Assessment: Thoroughness In Paper Reading:**

I read the paper thoroughly.

---

> ### Author Response · Authors · 2019-11-11
> **Response to Reviewer 3 (Part 1/2)**
>
> We would like to express our gratitude to the reviewer for giving many comments and also recognizing the strengths of our work, such as "The paper proposes to use residual learning as an auxiliary to compensate for the sub-optimal expressiveness of the source policies, which is novel and interesting."
>
> >>> Q: “In my understanding, the paper assumes that the source and target domain shared the same task (reward structure) but only differs in dynamics. Also, the two domains share similar state and action spaces since the policy accepts the target states and predict actions in the target environment. This may limit the usage of the method.” "One of my major concerns is that the limitation of the method with the same state and action space of the source and target domains"
>
> A: We would respectfully disagree with the reviewer's concern that our problem setting, which assumes source and target environments to be different only in their dynamics, limits the usage of our method. In fact, this problem setting is recognized as one of the common challenges in transfer RL, as introduced in a transfer RL survey [1], and has been studied extensively as we introduced briefly in Section 5 [2,3,4,5,6,7,8]. For instance, Yu et al. [8] presented their work in the last ICLR, which attempted to transfer policies between simulated and real robotic agents with different dynamics (https://openreview.net/pdf?id=H1g6osRcFQ), which we believe is a promising direction. Our work has advanced this line of research by proposing a new task and solution that could leverage multiple source policies without getting access to source environments. This contribution is recognized and supported by Reviewer 2: "an innovative contribution that pushes the needle on the transfer RL literature."
>
> Besides, our assumptions do not limit the real-world applicability of our method. For example, we could apply MULTIPOLAR to sim-to-real tasks [9], industrial insertion tasks [10] (different dynamics comes from the differences in parts), and wearable robots (with different users) [11].
>
> Having said that, we observe that our current introduction may not be sufficient to show how this problem setting is common and promising in the transfer RL domain, which leads to the reviewer's concern. We promise to clarify this point in the final version of our paper.
>
>
> [1] Matthew E. Taylor and Peter Stone. Transfer Learning for Reinforcement Learning Domains: A Survey. Journal of Machine Learning Research, 10(Jul):1633–1685, 2009.
> [2] Alessandro Lazaric, Marcello Restelli, and Andrea Bonarini. Transfer of Samples in Batch Reinforcement Learning. In International Conference on Machine Learning, pp. 544–551, 2008.
> [3] Haitham Bou Ammar, Eric Eaton, Matthew Taylor, Decebal Constantin Mocanu, Kurt Driessens, Gerhard Weiss, and Karl Tuyls. An Automated Measure of MDP Similarity for Transfer in Reinforcement Learning. In Workshops at the AAAI Conference on Artificial Intelligence, 2014.
> [4] Jinhua Song, Yang Gao, Hao Wang, and Bo An. Measuring the Distance Between Finite Markov Decision Processes. In International Conference on Autonomous Agents and Multiagent Systems, pp. 468–476, 2016.
> [5] Andrea Tirinzoni, Andrea Sessa, Matteo Pirotta, and Marcello Restelli. Importance Weighted Transfer of Samples in Reinforcement Learning. In International Conference on Machine Learning, pp. 4936–4945, 2018.
> [6] Tao Chen, Adithyavairavan Murali, and Abhinav Gupta. Hardware Conditioned Policies for MultiRobot Transfer Learning. In Advances in Neural Information Processing Systems, pp. 9355–9366, 2018.
> [7] Hao Wang, Shaokang Dong, and Ling Shao. Measuring Structural Similarities in Finite MDPs. In International Joint Conference on Artificial Intelligence, pp. 3684–3690, 2019.
> [8] Wenhao Yu, C. Karen Liu, and Greg Turk. Policy Transfer with Strategy Optimization. In International Conference on Learning Representations, 2019.
> [9] Jie Tan, Tingnan Zhang, Erwin Coumans, Atil Iscen1, Yunfei Bai, Danijar Hafner, Steven Bohez, and Vincent Vanhoucke. Sim-to-Real: Learning Agile Locomotion For Quadruped Robots. In Robotics: Science and Systems, 2018
> [10] Gerrit Schoettler, Ashvin Nair, Jianlan Luo, Shikhar Bahl, Juan Aparicio Ojea, Eugen Solowjow, Sergey Levin. Deep Reinforcement Learning for Industrial Insertion Tasks with Visual Inputs and Natural Reward Signals. In International Conference on Machine Learning Workshop, 2019
> URL: https://openreview.net/pdf?id=ryg5E-gy3E
> [11] Juanjuan Zhang, Pieter Fiers, Kirby A. Witte, Rachel W. Jackson, Katherine L. Poggensee, Christopher G. Atkeson, Steven H. Collins. Human-in-the-loop optimization of exoskeleton assistance during walking. Science 356.6344, pp. 1280-1284, 2017.

---

> > ### Author Response · Authors · 2019-11-11
> > **Response to Reviewer 3 (Part 2/2)**
> >
> > >>> Q: "the source and target domains only vary in some parameters of the agents. The domain gap seems small for these experiments." “The paper needs a metric to measure the domain gap between source and target dynamics and report how the domain gap influences the proposed method and the baselines according to the metric.” "There is no theoretical or intuitive analysis of how large the domain gap can be. This problem can be impractical for real-world applications with restrict limitations.”
> >
> > A: We are aware that quantifying domain gaps is an important step in transfer learning and domain adaptation methods in supervised learning settings. However, it is not common to do so in the transfer RL domain [1]. This is mainly because source/target data samples in transfer RL are collected only by agents interacting with source/target environments, which means that there are not static datasets that can be used to measure the gap between source and target environmental dynamics.
> >
> > Moreover, analyzing "how the domain gap influences the proposed method and the baselines" is also a nontrivial problem. For instance, Ammar et al. propose a data-driven measurement called RBDist to measure the domain gap [3], but the paper only shows that the proposed metric could be an indicator to predict transfer performance with some limited and preliminary examples. It’s not obvious how such metrics would work for the various tasks that we did, and how it could measure gaps when multiple source environments exist.
> >
> > To intuitively demonstrate the domain gap between source and target environment instances, we would like to share some videos of environment instances that were used in our Ant experiments https://www.youtube.com/watch?v=3b0mGeT3sLo. As shown in the videos, source and target robotic ants have different dynamics (e.g., different leg lengths, and frictions), which makes our problem very challenging.
> >
> > Moreover, as explained in our appendix, we designed the range of the environment dynamics/kinematics parameters for Roboschool environments following [6], which tackles a challenging problem in transfer RL with real-world applications.
> >
> > That being said, we don't mean that measuring the gap between source and target environments is pointless, and we find it an interesting question that should be addressed in the future work, especially to transfer policies to real-world environments.
> >
> > [1] Matthew E. Taylor and Peter Stone. Transfer Learning for Reinforcement Learning Domains: A Survey. Journal of Machine Learning Research, 10(Jul):1633–1685, 2009.
> > [3] Haitham Bou Ammar, Eric Eaton, Matthew Taylor, Decebal Constantin Mocanu, Kurt Driessens, Gerhard Weiss, and Karl Tuyls. An Automated Measure of MDP Similarity for Transfer in Reinforcement Learning. In Workshops at the AAAI Conference on Artificial Intelligence, 2014.
> > [6] Tao Chen, Adithyavairavan Murali, and Abhinav Gupta. Hardware Conditioned Policies for MultiRobot Transfer Learning. In Advances in Neural Information Processing Systems, pp. 9355–9366, 2018.

---

### Official Review · AnonReviewer1 · 2019-10-22
**Official Blind Review #1**

**Rating:** 8

**Review:**

This paper presents a transfer reinforcement learning method that learns from existing source policies. The method aggregates deterministic actions produced by a collection of source policies to maximize expected return in the target environment. Unlike prior work it does not assume access to source environments nor source policy performance.

The method is intuitive and simple (simply a weighted sum over the actions of source policies). The paper is well-written in that it clearly explains the method and intuitions. The authors show results on a collection of different environments that include continuous and discrete action spaces. I appreciate the additional work put in to evaluate the distribution of performance. The method is well-ablated and addresses variants in which there is no reweighting and in which the residual is estimated independently of the state.

I have some questions regarding the experiments:

- In Table 1, do the authors have intuitions for why sometimes RPL is worse than MLP?
- I'd like to see results comparing MULTIPOLAR with only bad sources with a randomly initialized policy
- Given that source policies are needed for this to work, I'd like to see comparisons in which one continues to finetune an existing source policy. I know that the assumption here is that one does not have access to the internals of the source policies, but it would be nice to see how the performance compares.

My main concern has to do with the applicability of this method, since it seems to make strong assumptions on how different the domain dynamics are between source and target environments.

**Experience Assessment:**

I have published one or two papers in this area.

**Review Assessment: Checking Correctness Of Derivations And Theory:**

I assessed the sensibility of the derivations and theory.

**Review Assessment: Checking Correctness Of Experiments:**

I assessed the sensibility of the experiments.

**Review Assessment: Thoroughness In Paper Reading:**

I read the paper at least twice and used my best judgement in assessing the paper.

---

> ### Author Response · Authors · 2019-11-11
> **Response to Reviewer 1 (Part 1/2)**
>
> We genuinely thank the reviewer for the thorough, detailed, and insightful review, which also highlights the strengths of our work such as "Unlike prior work it does not assume access to source environments nor source policy performance", "clearly explains the method and intuitions", and "the additional work put in to evaluate the distribution of performance". We would be happy to incorporate any other suggestions the reviewer may have for our paper. Below is our response to the questions raised in the review:
>
> >>> Q: “In Table 1, do the authors have intuitions for why sometimes RPL is worse than MLP?”
>
> A: This a thoughtful and relevant question. First of all, we would like to clarify that, in our experiments, we extended the RPL method [1] to use a single source policy that was not trained or hand-engineered for a target environment dynamics. Although this is not the assumption of the original RPL papers (they assume the source to be designed to work on a target environment), we had to do so due to the lack of baseline methods for our new problem setting, where RPL is the most relevant approach presented recently. We observe this might however raise the reviewer’s confusion, and promise to clarify this point in the final version of our paper.
>
> In our experiments, given that RPL had only a single source policy that was selected randomly from a pool of candidate source policies with diverse performance (shown in Figure 5 of the appendix A.4), it is likely that, on average, the selected source policy had a too low performance. On the other hand, MULTIPOLAR(K=4) has four source policies, which makes it more likely to have high-performing source policies helping the exploration or providing strong baseline actions.
>
>
> ---
> >>> Q: “I'd like to see results comparing MULTIPOLAR with only bad sources with a randomly initialized policy”
>
> A: We evaluated MULTIPOLAR (K=4), where the sources are just randomly initialized policies, in the Hopper environment using our experimental procedure explained in section 4.1. Below is the bootstrap mean and 95% confidence bounds of average episodic rewards over various training samples for this experiment and MULTIPOLAR with four low-performing sources:
>
> ===============================================================================
> MULTIPOLAR (K=4)                    |        0.5M          |        1M         |        1.5M        |       2M
> --------------------------------------------------------------------------------------------------------------------------------
> 4 low performance [Table 3]            27 (26,27)        45 (44,47)        68 (66,71)        92  (88,95)
> 4 randomly initialized                        27 (26,28)        47 (45,49)        73 (70,76)        101 (96,106)
> ===============================================================================
>
> This result shows that the sample efficiency of MULTIPOLAR with low-performing source policies (i.e., source policies which had low final episodic rewards in their own environments) is almost the same as with randomly initialized source policies.
>
> In the final version of our paper, we will incorporate this experiment in the appendix due to the main text page limitation.
>
>
> [1] Tobias Johannink, Shikhar Bahl, Ashvin Nair, Jianlan Luo, Avinash Kumar, Matthias Loskyll, Juan Aparicio Ojea, Eugen Solowjow, and Sergey Levine. Residual Reinforcement Learning for Robot Control. In International Conference on Robotics and Automation, pp. 6023–6029, 2019.

---

> > ### Author Response · Authors · 2019-11-11
> > **Response to Reviewer 1 (Part 2/2)**
> >
> > >>> Q: “Given that source policies are needed for this to work, I'd like to see comparisons in which one continues to finetune an existing source policy. I know that the assumption here is that one does not have access to the internals of the source policies, but it would be nice to see how the performance compares.”
> >
> > A: This is also an intriguing question. Based on this request, we conducted an extra experiment that fine-tuned a randomly selected source policy in target environment instances. Following our experimental procedure, the mean of average episodic rewards (over 3 random seeds and 3 random choices of a single source policy to be fine tuned in 100 environment instances) in the Hopper environment are:
> >
> > ==========================================================================================
> > Policy                      |               0.5M               |               1M               |               1.5M               |               2M
> > --------------------------------------------------------------------------------------------------------------------------------------------------
> > MLP (FineTuned)          997 (945,1049)            1209 (1157,1261)         1338 (1286,1390)           1428 (1378,1479)
> > ==========================================================================================
> >
> > We observe that the fine-tuned policies achieved a very high sample efficacy because they were able to 'learn' to interact with a target environment appropriately from the very early episodes to get a high return. With that said, this result does not affect our main contributions and conclusions that the proposed MULTIPOLAR worked well in the settings where source policies are hand-engineered or "one does not have access to the internals of the source policies," as recognized by the reviewer. To further investigate the comparison, we also visualized the average learning curves of fine-tuning an existing source policy, MULTIPOLAR, RPL, and baseline MLP, which is available here: https://www.dropbox.com/s/zosq4pd0bfooykl/all.pdf?dl=0
> >
> > These results suggest that MULTIPOLAR slightly outperforms finetuning an existing source policy in terms of final episodic rewards while finetuning is significantly more sample efficient than MULTIPOLAR.
> >
> >
> > ---
> > >>> Q: “My main concern has to do with the applicability of this method, since it seems to make strong assumptions on how different the domain dynamics are between source and target environments.”
> >
> > As the reviewer concerns, our method assumes that the environment structure (state/action space) is similar between source and target environments, while dynamics/kinematics parameters are different. However, please note that due to the nonlinearity of the environment dynamics, even little parameter changes would make a completely different environment  [2]. For example, a slight difference in dynamics parameters severely affects the low-level torque control in the robotic environments [2]. Hence, even with our assumption, the problem is still challenging.
> >
> > To intuitively demonstrate the difficulties, we would like to share some videos of environment instances that were used in our Ant experiments https://www.youtube.com/watch?v=3b0mGeT3sLo which is also available in our code repository. As shown in the videos, source and target robotic ants have different dynamics (e.g., different leg lengths, and frictions), which makes our problem very challenging.
> >
> > Besides, our assumptions do not limit the real-world applicability of our method. For example, we could apply MULTIPOLAR to sim-to-real tasks [3], industrial insertion tasks [4] (different dynamics comes from the differences in parts), and wearable robots (with different users) [5].
> >
> > Lastly, as explained in our appendix, we designed the range of the environment dynamics/kinematics parameters for Roboschool environments following [2], which tackles a challenging problem in transfer RL with real-world applications.
> >
> >
> > [2] Tao Chen, Adithyavairavan Murali, and Abhinav Gupta. Hardware Conditioned Policies for MultiRobot Transfer Learning. In Advances in Neural Information Processing Systems, pp. 9355–9366, 2018.
> > [3] Jie Tan, Tingnan Zhang, Erwin Coumans, Atil Iscen1, Yunfei Bai, Danijar Hafner, Steven Bohez, and Vincent Vanhoucke. Sim-to-Real: Learning Agile Locomotion For Quadruped Robots. In Robotics: Science and Systems, 2018
> > [4] Gerrit Schoettler, Ashvin Nair, Jianlan Luo, Shikhar Bahl, Juan Aparicio Ojea, Eugen Solowjow, Sergey Levin. Deep Reinforcement Learning for Industrial Insertion Tasks with Visual Inputs and Natural Reward Signals. In International Conference on Machine Learning Workshop, 2019
> > URL: https://openreview.net/pdf?id=ryg5E-gy3E
> > [5] Juanjuan Zhang, Pieter Fiers, Kirby A. Witte, Rachel W. Jackson, Katherine L. Poggensee, Christopher G. Atkeson, Steven H. Collins. Human-in-the-loop optimization of exoskeleton assistance during walking. Science 356.6344, pp. 1280-1284, 2017.

---

> > > ### Comment · AnonReviewer1 · 2019-11-14
> > > **Thank you**
> > >
> > > Thank you authors for your detailed response. I am  my score to Accept. Good luck!

---

### Official Review · AnonReviewer2 · 2019-10-28
**Official Blind Review #2**

**Rating:** 6

**Review:**

In this paper authors propose a method for transfer reinforcement learning (RL). Specifically they are claiming that RL agents can transfer knowledge between each other about the environment dynamics. In order to showcase their approach they have come up with a new transfer RL task that makes use of some source policies trained under a diverse set of environment dynamics. Their key contributions to solve the task involve a decision aggregation framework that is able to build on top of relevant policies while suppressing irrelevant ones and an auxiliary network that predicts the residuals around the aggregated actions.

I recommend the paper to be accepted since they have an innovative contribution that pushes the needle on the transfer RL literature although I do not think the contribution is substantial. The set of experiments covers a wide range of different standard RL tasks and they provide enough evidence that the approach works. I find it interesting that they are able to extend the approach to the discrete action tasks.

I would however recommend providing more experimental results that provides evidence that the target policy can recover the right policy when the target environment dynamics is the same as one of the source environments.

**Experience Assessment:**

I do not know much about this area.

**Review Assessment: Checking Correctness Of Derivations And Theory:**

I assessed the sensibility of the derivations and theory.

**Review Assessment: Checking Correctness Of Experiments:**

I assessed the sensibility of the experiments.

**Review Assessment: Thoroughness In Paper Reading:**

I read the paper at least twice and used my best judgement in assessing the paper.

---

> ### Author Response · Authors · 2019-11-11
> **Response to Reviewer 2**
>
> We would like to thank the reviewer for providing us with valuable comments and for recognizing the significance of our work, such as "an innovative contribution that pushes the needle on the transfer RL literature", "The set of experiments covers a wide range of different standard RL tasks and they provide enough evidence that the approach works", and "able to extend the approach to the discrete action tasks." Below is our response to the question raised in the review:
>
> >>> Q: “I would however recommend providing more experimental results that provides evidence that the target policy can recover the right policy when the target environment dynamics is the same as one of the source environments.”
>
> A: We suppose that the question here is: in the case that the target environment dynamics is the same as one of the source environments, will MULTIPOLAR policy converge to a solution where $\theta_{agg} \approx 1$ for that source policy, $\theta_{agg} \approx 0$ for the rest of the source policies, and $ F_{aux}(s_t; \theta_{aux}) \approx 0$? In other words, does MULTIPOLAR essentially learn to mirror the output actions from the source policy that is obtained from the environment instance with the same dynamics as the target?
>
> If we understood the question correctly, we would like to mention that, during the early stages of our work, we conducted several preliminary experiments with the suggested setup. We confirmed that MULTIPOLAR could learn a high-performing (sometimes significantly better than the source) policy very quickly; however, this was not achieved by recovering one of the source policies. The reason is twofold:
>
> (1) As discussed in our paper and shown in Figure 3, source policies are not guaranteed to work *optimally* in their environment instance. Hence, MULTIPOLAR converged to a better performing policy quickly by learning to aggregate all the source policies' actions and the residuals around them, rather than learning to recover one of the source policies.
>
> (2) In the space of policies, there may exist more than one high-performing solution for a given MDP [1].  Therefore, MULTIPOLAR, even in the case of having high-performing source policies,  may converge to any of the high-performing solutions rather than recovering a high-performing source policy. In fact, converging to a policy where $F_{aux}(s_t; \theta_{aux}) \approx 0$ would be quite difficult because it requires learning a function which maps every input states to a $D$-dimensional zero vector. Figure 6 of the appendix shows an example of a high-performing solution, in which MULTIPOLAR suppress the two useless low-performing policies and emphasizes the two high-performing policies, as the training progresses.
>
> Please also note that in real-world settings, it rarely happens that the source and target environment instances have the same dynamics because even a tiny deviation in the dynamics/kinematics parameters makes a substantial difference in the environment state transition distribution [2].
>
>
> [1] Richard S. Sutton,  and Andrew G. Barto. Reinforcement learning: An introduction. MIT press, 1998.
> [2] Tao Chen, Adithyavairavan Murali, and Abhinav Gupta. "Hardware conditioned policies for multi-robot transfer learning." Advances in Neural Information Processing Systems. 2018.

---

### Author Response · Authors · 2019-11-14
**General Response**

We would like to thank all the three reviewers for their overall positive and constructive comments.  Based on their feedback, we have made the following modifications to our paper:

1. Revised the introduction to include the real-world applications of our method, which address some of the Reviewer 1 and 3 concerns.

2. Revised Appendix B.2 to include the additional experiment on MULTIPOLAR with randomly initialized policies, suggested by Reviewer 1.

3. Added the URL of video replays of source policies, as well as MULTIPOLAR vs. baseline MLP in the Ant environment. This would clarify how different the source and target tasks are and confirm that our problem setting is challenging, which addresses some of the Reviewer 3 concerns.

4. Fixed a few typos.

5. Sectionalized Appendix B.

We did not include our other new experiment (suggested by Reviewer 1) that fine-tuned a randomly selected source policy in target environment instances. This is because fine-tuning an existing policy contradicts our primary assumption of not having access to the internals of the source policies.

We would also like to point out that in our initial submission, we made our code available and explained all the experimental details in our manuscript, which makes it possible to reproduce our experiments.

Finally, please find the detailed responses to each of the reviewers below.

---

### Decision · Program_Chairs · 2019-12-19

**Decision:**

Reject

**Comment:**

The paper considers the case where policies have been learned in several environments - differing only according to their transition functions. The goal is to achieve a policy for another environment on the top of the former policies. The approach is based on learning a state-dependent combination (aggregation) of the former policies, together with a "residual policy". On the top of the aggregated + residual policies is defined a Gaussian distribution. The approach is validated in six OpenAI Gym environments. Lesion studies show that both the aggregation of several policies (the more the better, except for the computational cost) and the residual policy are beneficial.

Quite a few additional experiments have been conducted during the rebuttal period according to the reviewers' demands (impact of the quality of the initial policies; comparing to fine-tuning an existing source policy).

A key issue raised in the discussion concerns the difference between the sources and the target environment. It is understood that "even a small difference in the dynamics" can call for significantly different policies. Still, the point of bridging the reality gap seems to be not as close as the authors think, for training the aggregation and residual modules requires hundreds of thousands of time steps - which is an issue in real-world robotics.

I encourage the authors to pursue this promising line of research; the paper would be definitely very strong with a proof of concept on the sim-to-real transfer task.